# Neutrophil extracellular traps arm DC vaccination against NPM-mutant myeloproliferation

Claudio Tripodo[1†], Barbara Bassani[2†], Elena Jachetti[2], Valeria Cancila[1], Claudia Chiodoni[2], Paola Portararo[2], Laura Botti[2], Cesare Valenti[3], Milena Perrone[2], Maurilio Ponzoni[4], Patrizia Comoli[5], Mara Lecchi[6], Paolo Verderio[6], Antonio Curti[7], Mario P Colombo[2‡], Sabina Sangaletti[2*‡]

[1]Tumor Immunology Unit, Department of Health Sciences, University of Palermo, Palermo, Italy; [2]Department of Research, Fondazione IRCCS Istituto Nazionale Tumori, Milan, Italy; [3]Department of Mathematics and Informatics, University of Palermo, Palermo, Italy; [4]Pathology Unit, IRCCS San Raffaele Scientific Institute, Milan, Italy; [5]Cell Factory, Istituto di Ricovero e Cura a Carattere Scientifico Policlinico San Matteo, Pavia, Italy; [6]Bioinformatics and Biostatistics Unit, Department of Applied Research and Technological Development, Fondazione IRCCS Istituto Nazionale dei Tumori, Milan, Italy; [7]Department of Experimental, Diagnostic and Specialty Medicine – DIMES, Institute of Hematology "Seràgnoli", Bologna, Italy

*For correspondence:
sabina.sangaletti@istitutotumori.mi.it

†These authors also contributed equally to this work
‡These authors are equally senior contribution to this work

**Competing interest:** The authors declare that no competing interests exist.

**Abstract** Neutrophil extracellular traps (NETs) are web-like chromatin structures composed by dsDNA and histones, decorated with antimicrobial proteins. Their interaction with dendritic cells (DCs) allows DC activation and maturation toward presentation of NET-associated antigens. Differently from other types of cell death that imply protein denaturation, NETosis preserves the proteins localized onto the DNA threads for proper enzymatic activity and conformational status, including immunogenic epitopes. Besides neutrophils, leukemic cells can release extracellular traps displaying leukemia-associated antigens, prototypically mutant nucleophosmin (NPMc+) that upon mutation translocates from nucleolus to the cytoplasm localizing onto NET threads. We tested NPMc+ immunogenicity through a NET/DC vaccine to treat NPMc-driven myeloproliferation in transgenic and transplantable models. Vaccination with DC loaded with NPMc+ NET (NPMc+ NET/DC) reduced myeloproliferation in transgenic mice, favoring the development of antibodies to mutant NPMc and the induction of a CD8+ T-cell response. The efficacy of this vaccine was also tested in mixed NPMc/WT bone marrow (BM) chimeras in a competitive BM transplantation setting, where the NPMc+ NET/DC vaccination impaired the expansion of NPMc+ in favor of WT myeloid compartment. NPMc+ NET/DC vaccination also achieved control of an aggressive leukemia transduced with mutant NPMc, effectively inducing an antileukemia CD8 T-cell memory response.

## Editor's evaluation

These findings are timely and novel. NPM1-mutated acute myeloid leukemia (AML) is a frequent AML subtype for which new therapeutic approaches are needed. The data presented support the feasibility and the anti-leukemic efficacy of a dendritic cell (DC) vaccine armed with neutrophil extracellular traps (NETs) derived from NPM1-mutated myeloid cells. The new methods presented have important implications and will be of interest to a broad audience from immunology, inflammation and cancer fields.

## Introduction

Fifteen years ago, a scanning electron microscopy image from Volker Brinkmann showed for the first time spider web-like chromatin structures extruded by neutrophils, called neutrophil extracellular traps (NETs), to entrap fungi and bacteria (*Brinkmann et al., 2004*). Such extracellular chromatin is composed by DNA and histones also decorated with antimicrobial proteins like myeloperoxidase (MPO) and neutrophil elastase suggesting the hypothesis that eukaryotic chromatin evolved under the need of maintaining genome integrity while defending the organism (*Brinkmann and Zychlinsky, 2012*). Extracellular traps can be released by other innate immune cells including eosinophils, macrophages, and mast cells (*Goldmann and Medina, 2012*) and, as recently shown, by cells of the adaptive immunity like CD4 T-helper cells (*Costanza et al., 2019*). Additionally, since their discovery, NET has been associated with inflammatory and immune-mediated diseases like diabetes, arthritis, systemic vasculitis, and lupus erythematosus (*Jorch and Kubes, 2017*). In a previous study, we demonstrated that the adoption of a NET-based dendritic cell (DC) vaccination was able to break tolerance against neutrophil cytoplasmatic antigens and induce antineutrophil cytoplasmatic antibodies (ANCA)-associated autoimmunity. Indeed, NET is highly immunogenic by virtue of the immune adjuvant effect of DNA and its associated proteins, as well as for the sticky properties of the DNA thread that enable NET persistent interaction with DCs for efficient loading of antigens followed by DC maturation and migration to draining lymph nodes for cross-presentation (*Sangaletti et al., 2012*).

Also, myeloid transformed cells can extrude NET into the extracellular space to activate the contact system of coagulation (*Demers et al., 2012*) or to sustain myeloproliferation. We have recently described extracellular traps enrichment in bone marrow (BM) biopsies from *NPM1* mutant acute myeloid leukemia (AML) patients and, using an ad hoc transgenic mouse, we showed that NPM cytoplasmic compartmentalization allows mutant NPM (NPMc+) to be relocalized onto the NET threads, exerting alarmin functions (*Tripodo et al., 2017*). The colocalization of histones with NPMc+ is a supporting evidence of their direct release from the leukemic clone.

The immune system has the capacity to eradicate AML as shown by the graft-versus-leukemia effect that is obtained after allogeneic hematopoietic stem cell transplantation. Accordingly, ideal immune molecular targets of AML should be restricted to leukemic cells including stem cells and be part of leukemic development. Indeed, the leukemia-associated antigens that entered clinical trials as peptide vaccines, such as RHAMM, Proteinase 3, and Wilms' tumor antigen-1 (WT-1), possess these features (*Anguille et al., 2012*).

*NPM1* mutations are among the most frequent molecular alterations in AML, where they play a prognostic role (*Falini et al., 2005*). *NPM1* gene encodes for the nucleolar protein nucleophosmin that regulates the ARF-p53 tumor-suppressor pathway (*Colombo et al., 2011*). NPMc mutations cause the stable relocalization of the protein from nucleus to cytoplasm, an event that, per se, is sufficient to trigger AML (*Falini et al., 2007*). The improved overall survival of patients with NPMc+ AML has been possibly explained by a T-cell response against the mutant epitopes. The specific CD4[+] and CD8[+] T-cell response against these epitopes raised the possibility of exploiting such property to immunize NPMc+ patients to control MRD or during maintenance treatment (*Schneider et al., 2014*). In this context, NET directly released by leukemic cells could efficiently display tumor-specific associated antigens and be used as vehicles for new DC-based vaccines.

In this study, we investigated whether the AML-associated NPMc is immunogenic when became part of the NET threads and whether it can be adopted in vaccination strategies to control leukemia outgrowth.

## Results

### NPMc+ NET/DC immunization controls NPMc-driven myeloproliferation

The h-MRP8-*NPM1+* (NPMc+) transgenic mice develop a myeloproliferation with expansion of mature CD11b+ myeloid cells and Gr-1+c-Kit+ myeloblasts, without development of overt acute leukemia. Using NET from NPMc+ transgenic mice we previously demonstrated that mutant NPM can work as an alarmin when localized in the cytoplasm as part of the NET thread (*Swerdlow et al., 2016*), gaining immunogenicity. In this work, we tested the possibility of using NPMc+ NET in a DC-based vaccination strategy to control myeloproliferation of NPMc+ transgenic mice. To this end, mice were immunized with DC cocultured with NPMc+ NET or WT NET (*Figure 1A*). During coculture, DC became loaded

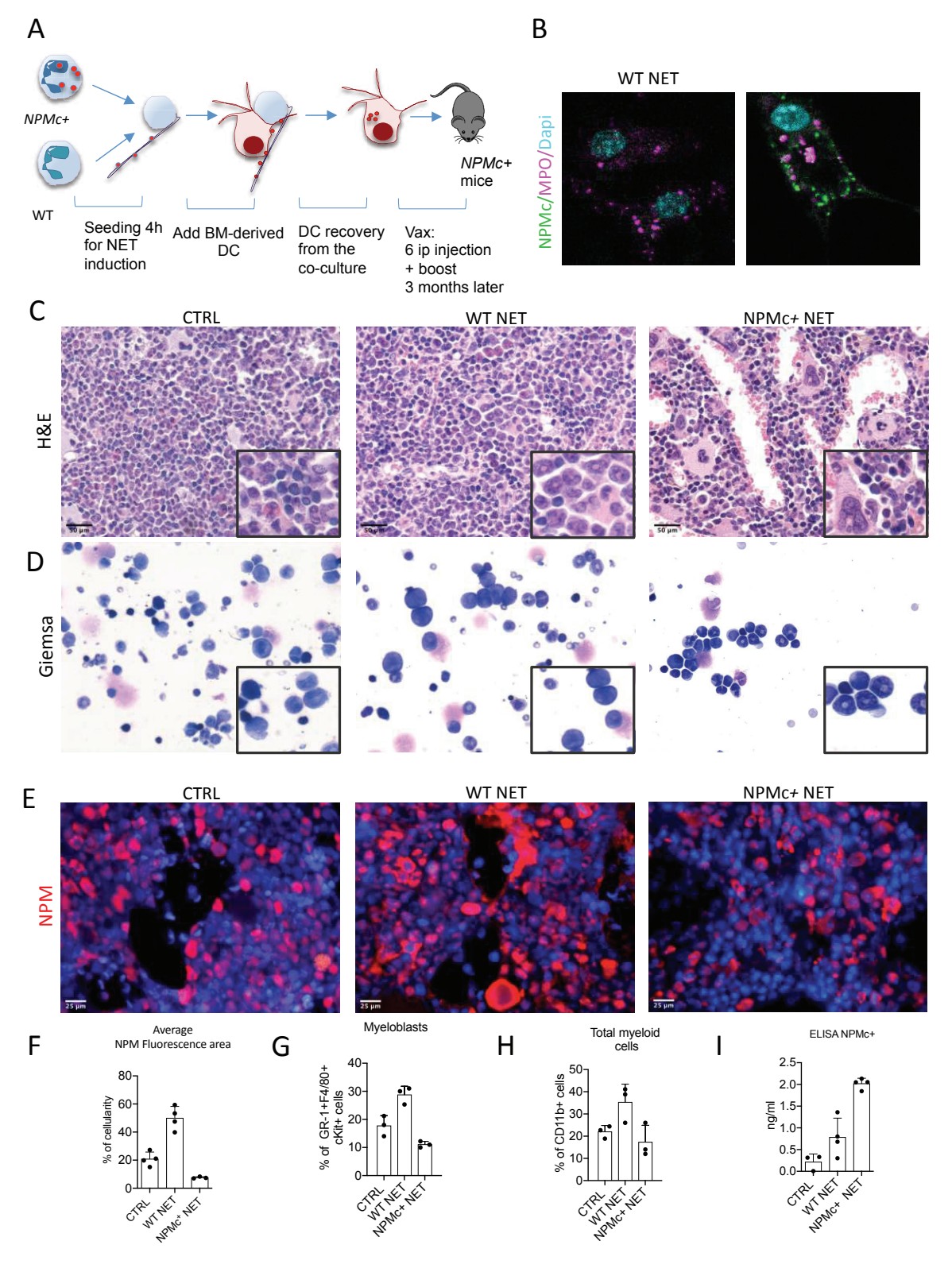

**Figure 1.** *Vaccination with NPM+ NET/DC controls NPMc+-driven myeloproliferation.* (**A**) Schematic representation of the vaccination experiment. (**B**) IF analysis for myeloperoxidase (MPO) (purple) and NPM (green) of DC cocultured with NPMc+ or WT NET. (**C**) Bone marrow (BM) histopathology of NPMc+ transgenic mice vaccinated with WT or NPMc+ NET-loaded DC or left untreated. (**D**) May–Grunwald Giemsa staining of BM smears from NPMc+ transgenic mice vaccinated with WT or NPMc+ NET-loaded DC or left untreated. (**E**) IF analysis for NPM on BM sections from vaccinated or

*Figure 1 continued on next page*

Figure 1 continued

control mice. (F) Quantification of NPMc + areas in the IF analysis (mean with SD; KW$_{MC}$ test p < 0.0001) (*Figure 1—source data 1*); PB FACS analysis for (**G**) GR-1+c-Kit+ myeloblasts (mean with SD; KW$_{MC}$ test p: 0.0030) and (**H**) CD11b+ myeloid cells (mean with SD; KW$_{MC}$ test p: 0.0810)(*Figure 1—source data 2*); NPMc+ transgenic mice vaccinated with WT or NPMc+ NET-loaded DC or left untreated. (I) Quantification of autoantibodies to mutant NPM developing in the serum of vaccinated mice (mean with SD; KW$_{MC}$ test p: 0.0021) (*Figure 1—source data 3*). In each graph every point represents a single mouse.

The online version of this article includes the following source data for figure 1:

**Source data 1.** Quantification of NPMc+ positive area.

**Source data 2.** FACS data relative to panels G and H.

**Source data 3.** ELISA data relative to the detection of NPM Ab in the sera of vaccinated mice.

with the prototypical NET-associated antigen MPO (*Figure 1B*, red signal) and, only in case of NPMc+ NET, with mutant NPMc (*Figure 1B*, green signal). NPMc+ NET/DC immunization, but not immunization with WT NET/DC, conspicuously reduced the signs of myeloid expansion in the BM of NPMc+ mice, with a decrease in dense clusters of morphologically immature granulocytic elements, as shown by histopathological analysis of H&E-stained BM sections (*Figure 1C*), and a reduction of myeloid blasts, in favor of more segmented forms on BM blood smears (*Figure 1D*). Moreover, NPMc+ NET immunization resulted in a significant decrease of cytoplasmic NPM-expressing elements, as assessed by immunofluorescence (*Figure 1E, F*, *Figure 1—source data 1*). Accordingly, Fluorescence Activated Cell Sorting (FACS) analysis of the PB confirmed the reduction of circulating immature GR-1+c-Kit+ precursors and of the overall frequency of CD11b+ cells (*Figure 1G, H*, *Figure 1—source data 2*) in the same vaccinated mice.

The immune response triggered by NPMc+ NET immunization induced anti-NPMc+ serum antibodies detected by ELISA (*Figure 1I*, *Figure 1—source data 3*) and the increase of CD8$^+$ T-cells frequency in BM infiltrates (not shown). Of note, the increased infiltration of CD8$^+$ T cells led to a higher frequency of CD8$^+$ T cells closely contacting NPMc+ cells (*Figure 2A, B*, and *Figure 2—source data 1*). Overall, these data indicate that DC vaccination with NPMc+ NET is able to induce immune activation toward control of NPMc+ myeloproliferation.

## NPMc+ NET/DC immunization selectively impairs NPMc+ mutant hematopoiesis in competitive BMT setting

To test the activity of the DC/NPMc+ NET vaccination in controlling the expansion of NPMc+ cells, we performed competitive BM transplantation experiments in which WT mice were transplanted with a 1:1 mixture of Lin- precursors from NPMc+ Tg mice (CD45.1) and WT mice (CD45.2). Four weeks after BMT mice were vaccinated with DC loaded with either WT or NPMc+ NET (*Figure 3A*). Mice were sacrificed 12 weeks after BMT and analyzed for the frequency of circulating CD45.1 (NPMc+ derived) and CD45.2 (WT-derived) CD11b+ and GR-1+c-Kit+ myeloblasts, respectively. Representative analysis (*Figure 3B*) and cumulative data (*Figure 3C* and *Figure 3—source data 1*) show that vaccination with NPMc+ NET-loaded DC impaired the expansion of CD45.1 mutant hematopoietic cells in favor of CD45.2 WT cells in the competitive setting. These results suggest that vaccination with NET carrying mutant NPMc could selectively control the expansion of mutant cells derived from NPMc+ precursors, while sparing the normal counterpart. Interestingly, the reduced expansion of CD45.1 NPMc+ cells was associated to increased CD8 T-cell infiltration in vaccinated mice (*Figure 3D, E* and *Figure 3— source data 2*). The selective effect on mutant cells in the competitive setting underscores the efficacy and specificity of the engendered immune response toward mutant hematopoiesis.

## NPMc+ NET/DC vaccination prevents transplantable NPMc+ leukemia ell growth

Since the indolent phenotype of NPMc+ Tg mice did not allow a short-term readout for assessing vaccine efficacy, we generated a transplantable leukemia model expressing mutant NPMc. To this purpose, the leukemia cell line C1498 was infected with a lentiviral vector expressing human mutant *NPM1* (C1498-NPMc+) (*Figure 4—figure supplement 1*) and then injected into NPMc+ Tg mice. In this experimental setting, the transgenic mice are expected to induce tolerance in case of potential transgene antigenicity (*Sow et al., 2020*), thus we tested whether NPMc+ NET-based vaccine

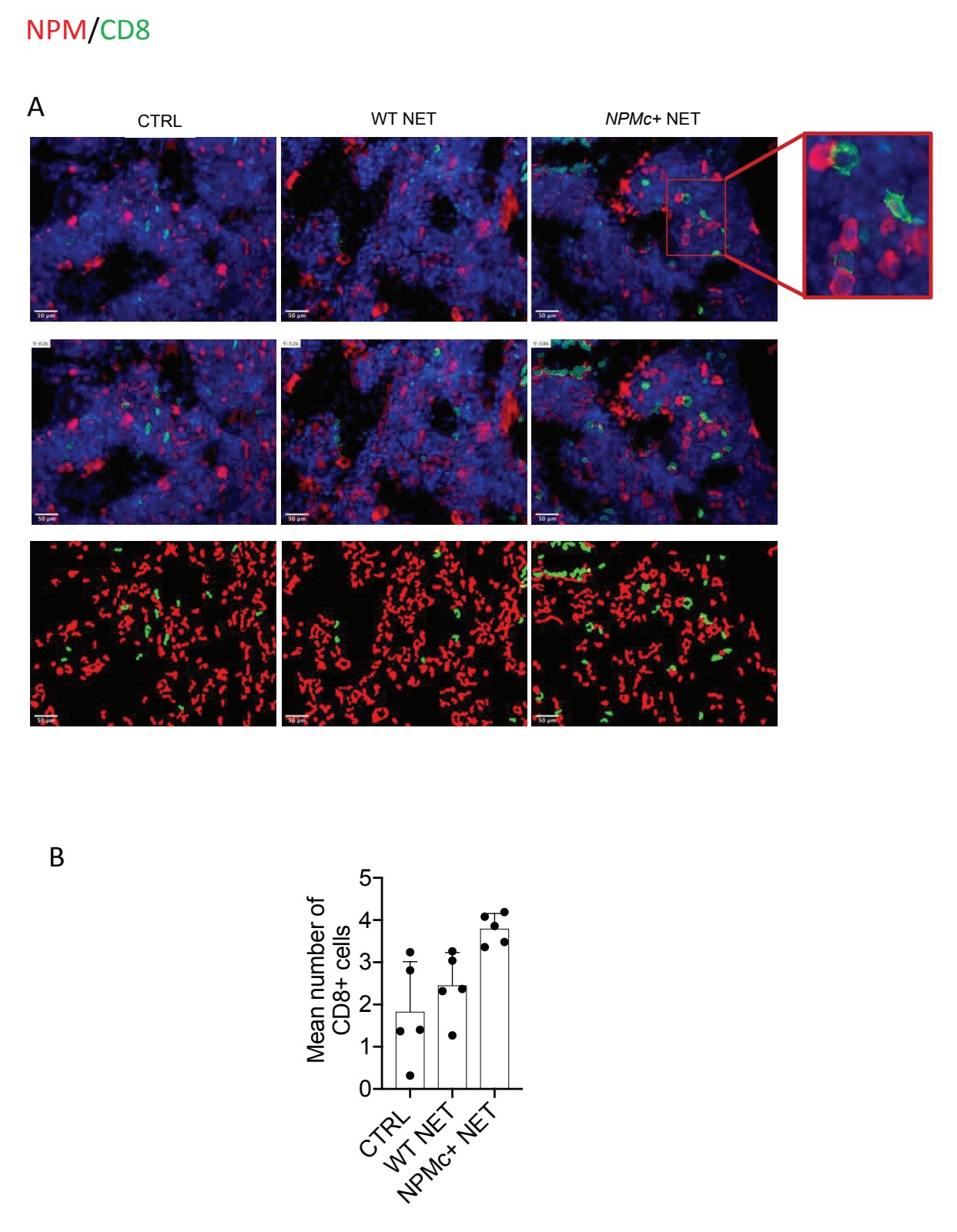

**Figure 2.** Analysis of CD8 T-cell frequency and interaction with NPMc+ cells in bone marrow (BM) biopsies from control and vaccinated mice. (**A**) Representative IF analysis on BM sections of NPMc+ transgenic mice vaccinated with WT or NPMc+ NET-loaded DC or left untreated, showing the reciprocal distribution of CD8+ T cells (green) and NPMc+ (red). (**B**) Software-based quantitative analysis of cell–cell contact between CD8+ and NPMc+ cells on segmented IF microphotographs; mean with SD, $KW_{MC}$ test p: 0.0016 (source data in *Figure 2—source data 1*).

*Figure 2 continued on next page*

*Figure 2 continued*

The online version of this article includes the following source data for figure 2:

**Source data 1.** Quantification of CD8/NPM contacts.

could disrupt tolerance in NPMc+ mice inducing an immune response able to control C1498-NPMc+ leukemia. DC uploaded with NPMc+ NET were injected intradermal in NPMc+ Tg mice bearing the C1498-NPMc+ leukemia. The vaccine was administered at days 3, 5, 10, and 14 post leukemia injection (*Figure 4—figure supplement 2A*). Tumor growth was monitored twice a week and the raising of NPMc-specific CD8+ T cells was evaluated through in vivo cytotoxicity assay (*Arranz et al., 2014*) by injecting mice with splenocytes pulsed 1 hr with NPMc-derived MHC-I peptides or with an unrelated peptide. The immunization significantly delayed tumor growth (*Figure 4A* and *Figure 4—source data 1*) and induced CD8+ T-cell cytotoxicity against NPMc (*Figure 4B* and *Figure 4—source data 2*). In a different arm of the same experiment, the NPMc +NET/DC vaccine also induced a strong CD8+ T cytotoxic response in tumor-free mice (*Figure 4B* and *Figure 4—source data 2*).

We next compared the effects of vaccination with DC loaded with either NPMc+ NET or NPMc-derived MHC-I-binding peptides QNYLFGCE, VEAKFINY, and LAVEEVSL. In a first set of experiments in which C1498-NPMc+ cells were injected subcutaneously, the NPMc+ NET/DC vaccine was superior to peptide-loaded DC in controlling tumor growth (*Figure 4C*, *Figure 4—source data 3*, *Figure 4—figure supplement 2B*). In situ IHC analysis revealed increased tumor-infiltrating CD8+ T cells and granzyme B+ elements in support of locally induced cytotoxic response, which was more robust in NPMc+ NET/DC than NPMc-derived MHC-I peptides/DC vaccinated mice (*Figure 4D–F* and *Figure 4—source data 4* and *Figure 4—source data 5*). To better mimic a physiologic condition of leukemia growth and to test the effects of our vaccines directly into the BM immune microenvironment, we performed intrabone injection of C1498-NPMc+ cells. The DC-based vaccines were administered ip at days 7, 12, 15, and 21. At end point, we evaluated the tumor take as the frequency of GFP+ leukemia cells in the BM. Results showed a reduction of GFP+ cells in the BM of mice vaccinated with either NPMc+ NET/DC vaccine or peptide/DC vaccine in comparison to nonvaccinated mice (*Figure 4G* and *Figure 4—source data 6*), which reached statistically significance in case of NPMc+ NET/DC vaccine (p = 0.0214 NPMc +NET/DC vs. ctrl and p = 0.0892 NPMc+ peptides vs. ctrl; Mann–Whitney *t*-test). Despite similar frequency of CD8 T cell (*Figure 4H*, *Figure 4—source data 7*), the NPMc+ NET/DC vaccine was superior in inducing the proliferation of CD8 T cells (*Figure 4I*, *Figure 4—source data 8*). Both vaccines sustained OX40 expression on CD8 T cells (*Figure 4J*, *Figure 4—source data 9*), however the reduction of exhausted PD1+TIM3+LAG3+ CD8 T cells (*Figure 4K*, *Figure 4—source data 10*) was higher in mice receiving the NPMc+ NET/DC vaccine. Accordingly, the production of TNF, but not IFNg, by CD8 T cells (*Figure 4L–M*, *Figure 4—source data 11*, and *Figure 4—source data 12*) was higher in mice receiving the NPMc+ NET/DC vaccine. Finally, a trend toward Tem increase was observed in mice receiving the NPMc+ NET/DC vaccine (*Figure 4N*, *Figure 4—source data 13*). ELISA assay showed that both vaccines were able to induce Auto-Ab against mutant NPMc+ (*Figure 4O*, *Figure 4—source data 14*), but only NPMc +NET/DC vaccinated mice developed Ab to MPO (*Figure 4P*, *Figure 4—source data 15*). Overall these data support the efficacy of NET-based vaccines that were for some aspects superior in promoting T-cell responses in comparison to canonical peptide-based vaccines.

## Discussion

Cancer immunotherapy can overcome the induction of drug resistance associated to standard treatments also establishing effective memory response that renders the therapeutic effect durable and independent of repetitive cycles of therapy. DC-loaded ex vivo with tumor antigens have largely been tested as cancer vaccines to induce Th1-type responses and trigger cytotoxic T cells targeting antigen-expressing tumor cells.

This strategy has been proposed for patients with AML, with variable clinical benefits (reviewed in *Van Acker et al., 2019*). Nevertheless, common weaknesses emerged from different clinical trials including the low ability of DC to mount an effective CD8 T-cell response because of a weak immunostimulatory activity and/or the existing immunosuppressive microenvironment instigated by leukemic clones fostering tolerogenic IDO-expressing DC (*Weinstock et al., 2017*).

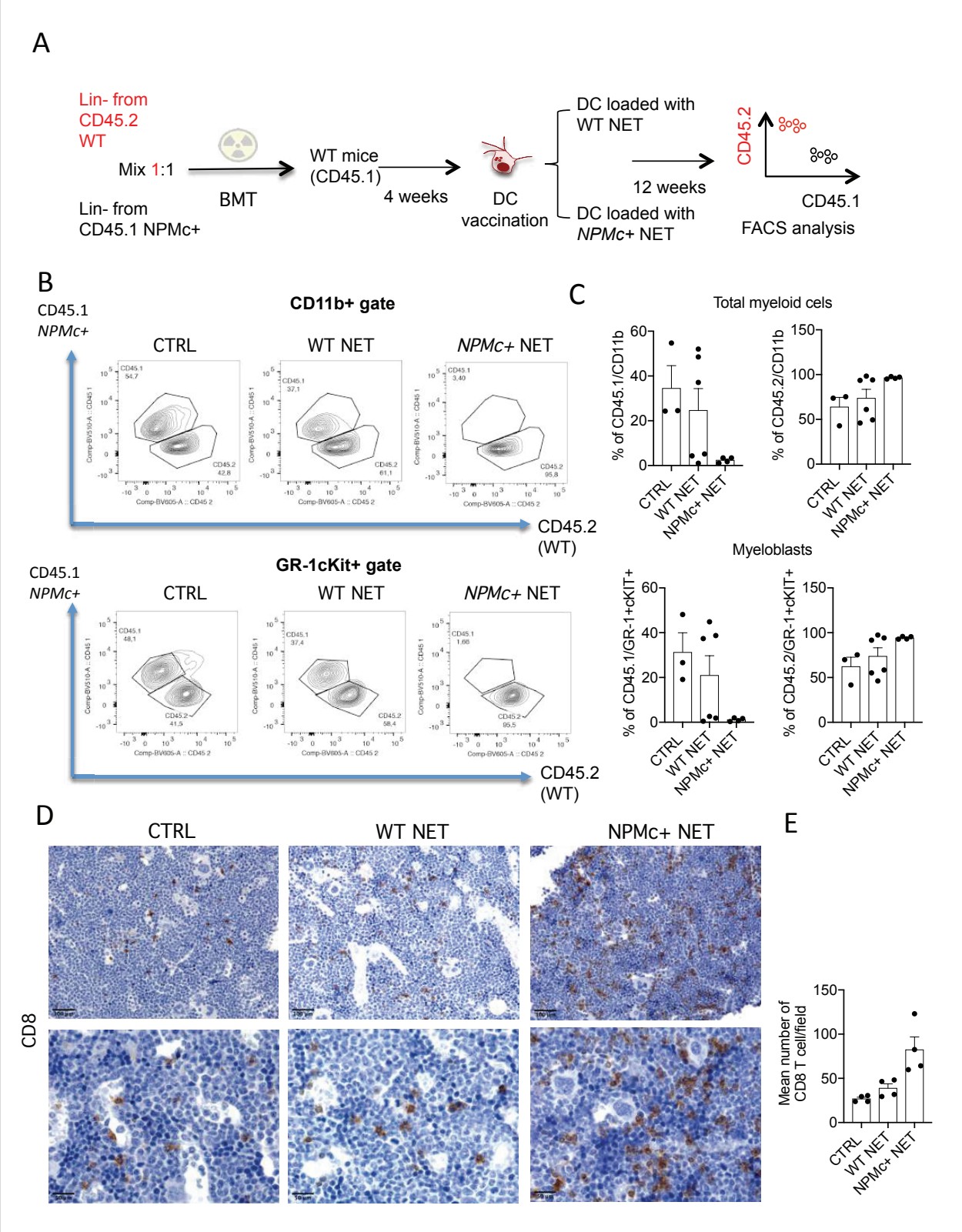

**Figure 3.** Vaccination with with NPM+ NET/DC controls the expansion of NPMc+ cells in competitive BMT assay. (**A**) Schematic representation of the competitive BMT experiment. (**B**) Representative dot plots showing the frequency of CD45.1 (NPMc+) and CD45.2 (WT) in myeloid cell- (CD11b+) and myeloblast- (GR-1+c-Kit+) gate of bone marrow (BM) chimeras that received vaccination with WT or NPMc+ NET/DC. (**C**) Cumulative data showing the frequency of CD45.1 (NPMc+) and CD45.2 (WT) within the CD11b+ (mean with SD; KW$_{MC}$ test p: 0.0628 and p: 0.0636, respectively) and GR-1+c-Kit+

*Figure 3 continued on next page*

*Figure 3 continued*

(mean with SD; KW$_{MC}$ test p: 0.0625 and p: 0.1134, respectively) gate (original data in *Figure 3—source data 1*). (**D**) IHC analysis of CD8$^+$ cells in BM sections of in BM chimeras vaccinated with WT or NPMc+ NET-loaded DC or left untreated. (**E**) Quantification of CD8$^+$ T cells performed by counting the number of immunoreactive cells out of five nonoverlapping high-power (×400) microscopic fields for every BM sample (mean with SD; KW$_{MC}$ test p: 0.0005) (original data in *Figure 3—source data 2*).

The online version of this article includes the following source data for figure 3:

**Source data 1.** Original FACS data for all the displayed groups and variables.

**Source data 2.** Quantification of infiltrating CD8$^+$ cells.

This work represents a proof-of-concept featuring NET-based DC vaccination of NPMc+ AML. This subset of AML embodies about 30% of total AML and 50%–60% of adult normal karyotype-AML and represents a distinct entity in the World Health Organization classification of Tumours of Haematopoietic and Lymphoid Tissues (*Swerdlow et al., 2016*). *NPM1*-mutant AML encompasses a spectrum of biological heterogeneity driven by the co-occurrence of genetic mutations (e.g., FLT3 internal tandem duplications) emerging at the transcriptional level (*Mer et al., 2021*).

NPMc+ AML has been characterized for spontaneous NPMc+ NET formation in the BM, an event associated with signs of immune activation (*Tripodo et al., 2017*). Despite the proimmune conditions endowed by a NET-permissive BM environment, the inflammatory skewing and the leukemic clone immunomodulatory profile associated with signs of myelomonocytic differentiation could eventually foster the development of a tolerogenic milieu suitable for NPMc+ AML progression, a condition potentially reverted by vaccination.

The main novelty of our work consists in the adoption of AML blast-derived NET as antigens source for DC presentation. This NET-based approach enables the display of leukemia-specific and -associated antigens, such as NPMc, and the conveyance of costimulatory signals through the dsDNA thread. In the case of NPMc+ AML, we demonstrated that mutant NPMc exerts a potent adjuvant function, similar to that described for the alarmin HMGB1 (*Tripodo et al., 2017*).

Our results show that NPMc+ NET/DC vaccination is able to break tolerance against mutated NPM, eventually promoting a strong cytotoxic activity in naive and in AML-bearing NPMc+ Tg mice. These experimental results could represent an important complement to the reported effect of anti-PD-1 immunotherapy in inducing antigen-specific cytotoxic T-cell responses against immunogenic epitopes derived from mutant *NPM1*, which indicates an existing antigen-specific CD8 response in NPMc+ AML patients (*Greiner et al., 2020*). Comparing NPMc+ NET/DC vaccines with DC pulsed with NPMc-derived peptides, we found a consistent capacity of NET-based vaccines to promote antitumor immunity inducing CD8 T-cell responses along with the development of NPMc-specific antibodies. Although the efficacy of the NET-based vaccine to control leukemia growth was similar to the DC+ peptides counterpart, only the former reach statistical significance for several immune parameters, in a side-by-side comparison. Also, we would stress the novelty of the NET-based vaccines in AML, using the leukemic blasts as source of NETs, which largely represent the majority of the AML antigenic repertoire displayed onto DNA threads, per se endowed of adjuvant functions. Differently, peptide-based vaccines require prior knowledge of the immunogenic peptides to be loaded onto in vitro-activated DCs.

One potential limitation of the proposed NET/DC vaccination strategy lies on the possibility of inducing autoimmune responses against NET-exposed self-antigens, as in the case of antineutrophil cytoplasmic antibodies induced against MPO or proteinase-3. Autoimmune vasculitides of small vessels have been detected in peripheral tissues as the result of NET-loaded DC intraperitoneal injection (*Sangaletti et al., 2012*), but no signs of autoimmunity were observed upon adoption of a subcutaneous NET/DC vaccination, as for the here presented vaccination of C1498-NPMc+ tumor-bearing transgenic mice. Another potential caveat is related to the long-term effects of the immune response elicited by NPMc+ NET/DC vaccination on the hematopoietic niche and on its sustaining nonmalignant hematopoiesis. Experiments with BM chimeras show that NPMc+ NET/DC vaccination selectively impaired the NPMc+ elements, without disrupting the WT hematopoiesis. However, if the induction of sustained immune responses targeting BM resident elements can in turn favor the establishment of proinflammatory conditions contributing to BM niche disfunction is hard to be envisaged, in patients. BM hematopoietic stromal niche alterations have been demonstrated to causally affect the arousal and progression of hematopoietic malignancies, either through transcriptional/phenotypical

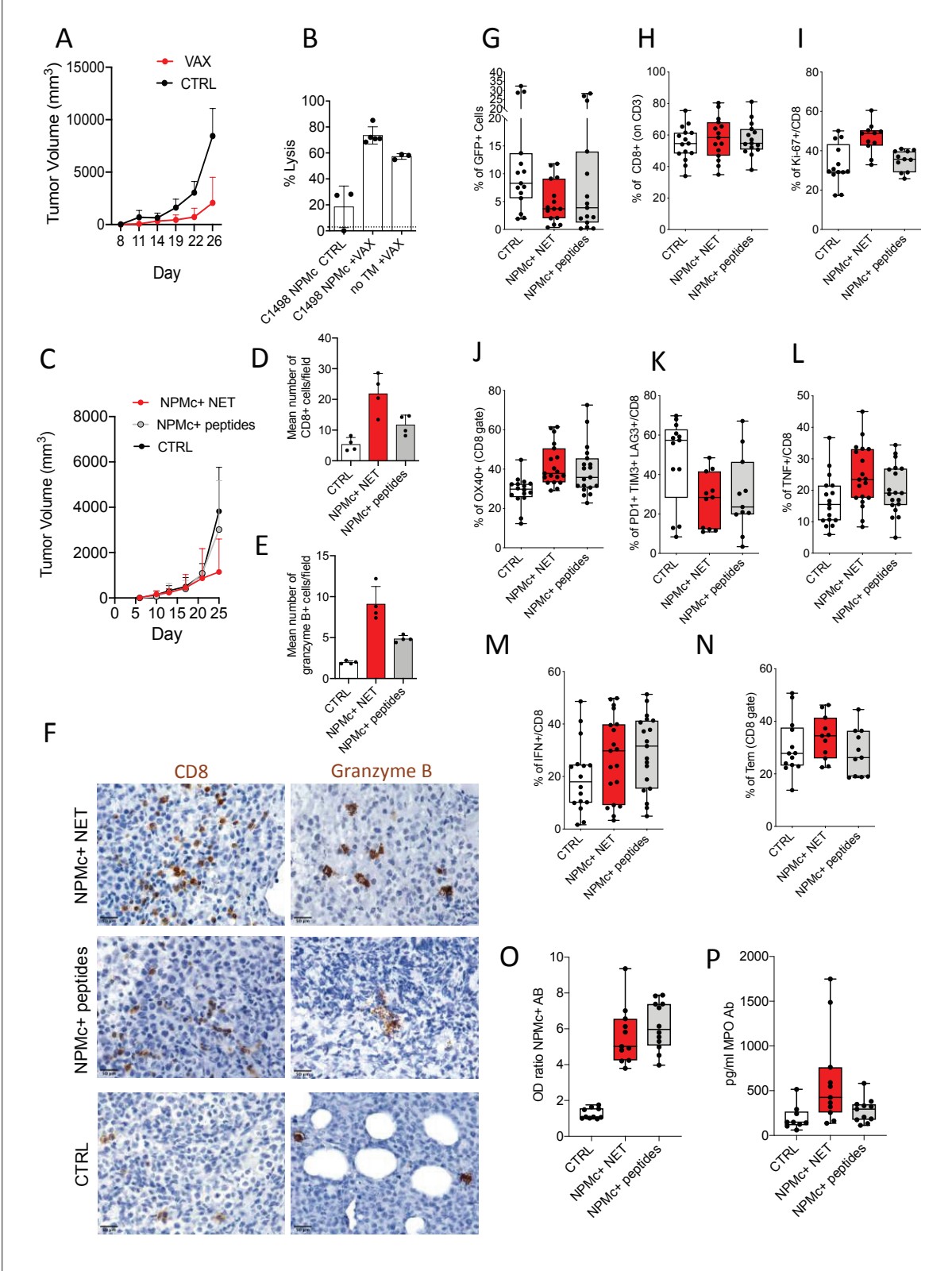

**Figure 4.** NPMc+ NET vaccination prevents transplantable NPMc+ leukemia cell growth and promotes CD8 lysis. C1498-NPMc+ leukemia cells were injected s.c. into NPMc+ transgenic mice. NPMc+ NET/DC-based vaccine was administered at days 3, 5, 10, and 14 post leukemia injection. Tumor growth was monitored twice a week. (A) Line chart of the mean with standard deviation (SD) tumor volume (mixed model, p: 0.0214) (original data in *Figure 4—source data 1*). (**B**) Elicitation of antigen-specific CD8+ T cells in vaccinated mice. Vaccinated tumor bearing mice (TB) or control mice

*Figure 4 continued on next page*

*Figure 4 continued*

have been injected with $10^7$ cells containing equal numbers of splenocytes labeled with 1.25 μM (CFSE$^{hi}$) or 0.125 μM of CFSE (CFSE$^{low}$). CFSE$^{hi}$ cells were previously pulsed 1 hr with NPMc-MHC-I peptides. Mice were sacrificed the following day, and their splenocytes and lymph nodes analyzed by flow cytometry for the evaluation of the presence of CFSE$^{hi}$ and CFSE$^{low}$ cells. NPMc-specific cytolytic activity was calculated as: (percentage CFSE$^{high}$ cells) × 100/(percentage CFSE$^{low}$ cells) (mean with SD; KW$_{MC}$ test p: 0.0007; dotted line refers to the control 'no TM CTRL' (*Figure 4—source data 2*). One representative experiment out of three performed. Abbreviations: No TM: mice noninjected with C1498 cells; VAX: vaccinated mice; CTRL: nonvaccinated mice. (**C**) Take of C1498-NPMc+ cells injected s.c. in NPMc+ transgenic mice vaccinated with DC pulsed with NPMc+ NET or NPMc+ peptides at days 3, 5, 10, and 14 post leukemia injection, line chart of the mean with SD tumor volume (mixed model, p: 0.1443). IHC analysis for CD8 and granzyme B of C1498 tumors subcutaneously grown in NPM1 tg mice that received the different vaccinations. (**D**) quantification of CD8 (mean with SD; KW$_{MC}$ test p: 0.0010) (original data in *Figure 4—source data 4*) and (**E**) granzyme B+ cells (mean with SD; KW$_{MC}$ test p: 0.0001) (original data in *Figure 4—source data 5*) and (**F**) representative pictures showing CD8 and granzyme B+ cells in tumors from mice vaccinated with NPMc+ NET or NPMc+ peptides. (**G**) Frequency of GFP+ cells in mice injected intrabone and vaccinated with DC pulsed with NPMc+ NET or NPMc+ peptides at days 10, 14, 17, and 23 post leukemia cell injection (KW test p: 0.064) (original data in *Figure 4—source data 6*). FACS analysis showing the frequency of CD8 T cells (KW test p: 0.7496) (**H**; original data in *Figure 4—source data 7*), Ki-67+ CD8 T cells (KW test p: 0.0028) (I, original data in *Figure 4—source data 8*), OX40 + CD8 T cells (analysis of Variance [ANOVA] two-way test, p:0.0066) (**J**, original data in *Figure 4—source data 9*), exhausted T cells (PD1+TIM3+LAG3+, panel K, original data in *Figure 4—source data 10*). KW test p: 0.0417, TNF+ (ANOVA two-way test, p: 0.0324) and IFNg+CD8 T cells (ANOVA two-way test, p: 0.4577) (**L, M**, original data in *Figure 4—source data 11* and *Figure 4—source data 12*) and the amount of effector memory cells (KW test p: 0.3011) (**N**, original data in *Figure 4—source data 13*). Titer of Ab to mutant NPMc and MPO in the sera of vaccinated mice. The NPMc Ab titer (**O**, original data in *Figure 4—source data 14*) is shown as OD ratio (KW test p: < 0.0001) whereas the MPO Ab titer (**P**, original data in *Figure 4—source data 15*) is shown as pg/ml (KW$_{MC}$ test p: 0.0092). Each boxplot (**G–P**) indicates the 25th and 75th centiles of the distribution. The horizontal line inside the box indicates the median and the whiskers indicate the extreme measured values.

The online version of this article includes the following source data and figure supplement(s) for figure 4:

**Source data 1.** Tumor growth evaluation in vaccinated mice.

**Source data 2.** FACS data relative to the in vivo cytotoxicity assay.

**Source data 3.** Tumor growth evaluation in mice receiving DC+NPMc+ NET or DC+NPMc+ peptides vaccines.

**Source data 4.** Quantification of infiltrating CD8$^+$ cells by IHC.

**Source data 5.** Quantification of infiltrating granzyme B+ cells.

**Source data 6.** Original FACS data relative to the frequency of GFP+ cells.

**Source data 7.** Original FACS data relative to the frequency of CD8$^+$ cells (CD3 gate).

**Source data 8.** Original FACS data relative to the frequency of Ki-67+/CD8 cells.

**Source data 9.** Original FACS data relative to the frequency of OX40+ cells (CD8 gate).

**Source data 10.** Original FACS data relative to the frequency of PD1+TIM3+LAG3+ on CD8 cells.

**Source data 11.** Original FACS data relative to the frequency of TNF+ cells (on CD8+).

**Source data 12.** Original FACS data relative to the frequency of IFNg+ cells (on CD8+).

**Source data 13.** Original FACS data relative to the frequency of Tem (on CD8+).

**Source data 14.** ELISA assay for Ab to mutant NPM original data.

**Source data 15.** ELISA assay for Ab to mutant myeloperoxidase (MPO) original data and titration curve.

**Figure supplement 1.** Detection of mutant NPMc+ in C1498 cells infected with a lentiviral vector expressing human mutant NPM1.

**Figure supplement 2.** NPMc+ NET vaccination prevents transplantable NPMc+ leukemia cell growth.

rewiring of key cellular components of the niche (e.g., osteoblasts, nerve fibers, mesenchymal stromal cells) (*Kode et al., 2014*; *Arranz et al., 2014*; *Zambetti et al., 2016*) or through the modulation of extracellular matrix organization (*Tripodo et al., 2012*). In this regard, extracellular matrix amount and composition have been demonstrated to affect the activation state, cross-presentation activity, and immunoregulatory properties of myeloid elements, an issue that could prove of particular relevance in the unfavorable setting of myeloid malignancies associated with BM fibrosis (*Sangaletti et al., 2005*; *Sangaletti et al., 2016*).

## Materials and methods
### Animals
hMPR8-*NPM1* transgenic mice, hereafter referred as *NPM1* tg mice, have been acquired on a mixed (C57BL/6J × CBA) F1 background (*Tripodo et al., 2017*) and backcrossed to C57BL/6 (B6.Ptprc <

a > ) mice. All experiments involving animals described in this study were approved by the Ministry of Health (authorization number 443/2016-PR e 693/2018-PR). NPMc+–C1498 cells were obtained through lentiviral infection of C1498 cells that were provided by ATCC. NPMc+–C1498 cells were authenticated by Microsynth service using highly polymorphic short tandem repeat loci (STRs). Cell lines are routinely checked for Mycoplasma negativity using MycoAlert Mycoplasma detection Kitt (Lonza). For the intrabone injection mice received $2 \times 10^5$ cells in 20 µl of saline into the tibia bone cavity. For the subcutaneous injection mice received $5 \times 10^5$ cells in 200 µl of saline into the right flank.

## Lentiviral vector construction, virus production, and cell infection

To construct *NPM1* lentivector the mutant *NPM1* cDNA (mutation type A, TCTG duplication) was cloned into the pLVX-EF1α-IRES-ZsGreen vector (Clontech) using EcoRI and BamHI restriction sites.

A third-generation packaging system was used to produce viral particles. Lentiviral supernatants for *NPM1*-expressing virus and control virus (empty pLVX-EF1α-IRES-ZsGreen vector) were produced in 293T cells by Ca3PO4 cotrasfection of the plasmids as previously described (*De Palma and Naldini, 2002*). C1498 cell line was infected using viral supernatants at 1:2 ratio with RPMI complete medium. Percentage of infection was evaluated by flow cytometry for GFP expression. Subsequently, infected C1498 cells were sorted by FACSAria to obtain pure populations (100% GFP expression).

## Generation of mouse BM chimeras

Competitive BM transplantation assay has been performed transplanting lethally irradiated (1100 RAD) WT mice (B6.Ptprc < a > ; CD45.1) with $2 \times 10^5$ Lin negative cells from either *NPMc+* (B6.Ptprc < a > ;CD45.1) or WT controls (B6.Ptprc < b > ; CD45.2). LK cells were isolated from total BM through negative selection (Lineage Cell Depletion Kit, Miltenyi). This method allows achieving 70% of purity rate.

## NET-based vaccination protocol

Neutrophils prone to extrude traps have been isolated from agar plugs implanted subcutaneously in WT and *NPM1* transgenic mice, as previously described (*Sangaletti et al., 2012*). In particular, agar NET-prone PMN were seeded onto coated tissue culture dishes in Iscov's Modified Dulbecco's Medium ( IMDM) 2% FCS, allowed to adhere for 30 min and added with myeloid DC (1:1; PMN:DC) for 16 hr. During this period, NET is induced and transfer their component to DC that in turn upregulate MHC-II and costimulatory molecules (*Sangaletti et al., 2012*). Then DC are isolated from the coculture via positive selection, counted and injected (Miltenyi, CD11c Microbeads Ultra Pure). We used three schedules for the administration of the DC NET-based vaccines: (1) ip injection: $2.5 \times 10^6$ cells, once a week for a total of six injection followed by a boost 3 months later; (*Figure 1A*) or (2) intradermal: $2 \times 10^6$ cells at days 3, 5, 10, and 14 post leukemia cell (C1498 cells) injection, (Schedule in *Figure 4A*); ip injection at days 7, 12, 15, and 21 for mice receiving C1498 cells intrabone (*Figure 4G*).

For peptide-based vaccination protocol, BM-derived DC have been treated with Lipopolysaccharide (LPS) (10 ng/ml) for 16 hr and then added with a mixture of three *NPMc*+MHC-I-binding synthetic peptides for 2 hr. DC were washed and used in vaccination experiments. MHC-I H2-Kb peptides for *NPM1*-A mutant NPMc protein were designed according to the best score interrogating the epitope prediction site syfpeithi.de. Peptides with the following amino acid sequences, QNYLFGCE, VEAKFINY, and LAVEEVSL, were purchased from Primm Biotech (Milan, Italy). Peptides were prepared in Dimethyl sulfoxide (DMSO) and then diluted in distilled water.

## Flow cytometry

Staining for cell surface markers was performed in phosphate-buffered saline (PBS) supplemented with 2% fetal bovine serum (FBS) for 30 min on ice. Flow cytometry data were acquired on a LSRFortessa (Becton Dickinson) and analyzed with FlowJo software (version 8.8.6 and 10.4.2, Tree Star Inc). To assess myeloproliferation in *NPMc*+ transgenic mice, emphasis has been given to the evaluation of the expansion of myeloid precursors. These cells have been identified tanks to their coexpression of CD11b, GR-1, and c-Kit+ markers as described (*Colombo et al., 2011*). Memory CD4 and CD8 T cell have been identified within their respective gates according to the expression of CD44 and CD62L. Megakaryocyte precursors have been identified within the Lin-c-Kit- gate according to their

coexpression of CD41 and CD150 markers. All the antibodies that have been used in flow cytometry are listed in Table S1.

## In vivo cytotoxicity assays

Mice were immunized as described and euthanized 1 week after the last boost. In vivo cytotoxicity assay was performed as described in *Degl'Innocenti et al., 2005*. Briefly, the day before sacrifice, mice were injected i.p. with $10^7$ cells containing equal numbers of splenocytes labeled with 1.25 mg/ml (CFSE$^{hi}$) or 0.125 mg/ml of CFSE (CFSE$^{low}$). CFSE$^{hi}$ cells were previously pulsed 1 hr with a mixture of three different NPMc-derived peptides (peptide sequences: QNYLFGCE; VEAKFINY; LAVEEVSL). Upon sacrifice, splenocytes were analyzed by flow cytometry for the presence of CFSE$^{hi}$ and CFSE$^{low}$ cells. NPMc-specific cytolytic activity was calculated as: (percentage of CFSE$^{high}$ cells) × 100/(percentage of CFSE$^{low}$ cells).

## Preparation of bone smears

BM was flushed with saline from the long bones with a 1-ml sirynge in a small plate then recovered with the help of forceps and then smeared onto a slide.

## Histopathological analysis, immunohistochemistry, and immunofluorescence

To assess myeloproliferation, the BM of NPMc Tg vaccinated or control mice has been histopathologically evaluated for cellularity, expansion of myeloid cells and degree of maturation of myeloid components. Histopathological analysis was performed on routinely stained hematoxylin-and-eosin sections by a pathologist with specific expertise in hematopathology and murine pathology (CT). Briefly, tissue samples were fixed in 10% buffered formalin, decalcified using an Ethylenediaminetetraacetic Acid (EDTA)-based decalcifying solution (MicroDec, Diapath) and paraffin embedded. Four-micrometers-thick sections were deparaffinized and rehydrated. The antigen unmasking technique was performed using Novocastra Epitope Retrieval Solutions pH 6 and pH 9 in a thermostatic bath at 98°C for 30 min. Subsequently, the sections were brought to room temperature and washed in PBS. After neutralization of the endogenous peroxidase with 3% $H_2O_2$ and Fc blocking by a specific protein block (Novocastra, UK), the samples were incubated with the primary antibodies. The following primary antibodies were adopted for IHC and IF: NPM (clone 376, dilution 1:100 pH 6, Dako); CD8α (clone D4W2Z, dilution 1:400 pH 9, Cell Signaling); and granzyme B (dilution 1:10 pH 6, Cell Marque). The staining was revealed using IgG (H&L) specific secondary antibody HRP-conjugated (1:500, Novex by Life Technologies) and DAB (3,3'-diaminobenzidine, Novocastra) as substrate chromogen. For double-marker immunofluorescence, after antigen retrieval, the sections were incubated overnight at 4°C with NPM and CD8α primary antibodies. The binding of the primary antibodies to their respective antigenic substrates was revealed by made-specific secondary antibodies conjugated with Alexa-488 (Life Technologies, 1:250) and Alexa-568 (Life Technologies, 1:300) fluorochromes. Nuclei were counterstained with DAPI (4',6-diamidin-2-fenilindolo). Slides were analyzed under an Axioscope A1 microscope equipped with an Axiocam 503 Color digital camera and Zen 2.0 Software (Zeiss). Quantitative IHC data were obtained by calculating the number of CD8$^+$ or granzyme B+ cells in five nonoverlapping fields at high-power magnification (×400).

To measure the cell–cell contact between NPMc+ cells (green signal) and CD8$^+$ cells (red signal), we applied the redundant wavelet transform called à trous (*Sciortino et al., 2017*; *Bellavia et al., 2014*) because it preserves the original resolution of the image while allowing the removal of noise. With such a method, the quality of the data was preserved without the need of pre- or postprocessing techniques. The intensity of overlay signal (yellow signal) was then measured only within the significant zones identified automatically and related to their own areas. This approach makes the methodology we developed independent of the zoom factor used to acquire the photomicrographs.

## Confocal microscopy analysis

To evaluate NET formation, NPMc relocalization onto the NET threads, and the transfer of MPO and NPMc+ antigens to DC, agar neutrophils from WT and NPMc+ mice were seeded onto poly-D-lysine coated glasses (IMDM 2% FCS), allowed to adhere for 30 minutes and added with mDCs (1:1 PMN:DC). After 16 h, cells were fixed in 4% paraformaldehyde (PFA) and sequentially stained with

mAbs to NPM (antihuman/mouse NPM [clone 376, dilution 1:50, Dako] and antimouse MPO [Millipore]). The NET-DNA was counterstained with Draq5 or the vital DNA dye Sytox green, or with DAPI. IF stainings were acquired under a Leica TCS SP8 confocal microscope (Leica Microsystems).

## Statistical analysis

The comparisons of interest were assessed by resorting to the nonparametric Kruskal–Wallis (KW) test. p values were estimated via Monte Carlo ($KW_{MC}$) approach when appropriate. Two-way analysis of variance was adopted to jointly consider observations arisen from experiments performed at different times (block variable). Mixed models (with a compound symmetry covariance matrix) were fitted to assess the tumor growth as a function of time and experimental group (fixed factors) with mice considered as random factor.

All statistical analyses were carried out with SAS (Statistical Analysis System, RRID:SCR_008567, version 9.4; SAS Institute, Inc, Cary, NC, USA), adopting an α level of 5%.

Graphical representations of dotplots reporting mean with standard deviation or boxplots were obtained with Prism version 9.2 (GraphPad Software, San Diego, CA, USA).

## Acknowledgements

This work has been supported by the Italian Ministry of Health (GR-2013-02355637 to S Sangaletti), Fondazione AIRC per la Ricerca sul Cancro (AIRC) (IG 22204 to Sabina Sangaletti, IG 24363 to Mario P Colombo and 5x1000 program ID 22759 group leader Claudio Tripodo). B Bassani is funded by the Fondazione Italiana per la Ricerca sul Cancro-Fondazione AIRC per la Ricerca sul Cancro (FIRC-AIRC) fellowship 'Guglielmina Lucatello e Gino Mazzega'. The authors are grateful to Ester Grande for administrative support.

## Additional information

### Funding

| Funder | Grant reference number | Author |
| --- | --- | --- |
| Fondazione AIRC per la Ricerca sul Cancro | IG22204 | Sabina Sangaletti |
| Ministry of Health | GR-2013-02355637 | Sabina Sangaletti |
| Fondazione AIRC per la Ricerca sul Cancro | 22369 | Barbara Bassani |
| Fondazione AIRC per la Ricerca sul Cancro | 5x1000 program, grant ID22759 | Claudio Tripodo |
| Fondazione AIRC per la Ricerca sul Cancro | IG24363 | Mario P Colombo |

The funders had no role in study design, data collection, and interpretation, or the decision to submit the work for publication.

### Author contributions

Claudio Tripodo, Formal analysis, Funding acquisition, Investigation; Barbara Bassani, Investigation, Formal analysis, Funding acquisition, Writing – original draft; Elena Jachetti, Investigation, Formal analysis; Valeria Cancila, Paola Portararo, Laura Botti, Methodology; Claudia Chiodoni, Investigation, Methodology; Cesare Valenti, Methodology, Investigation; Milena Perrone, Investigation; Maurilio Ponzoni, Patrizia Comoli, Antonio Curti, Writing – review and editing; Mara Lecchi, Methodology, Statistical analysis; Paolo Verderio, Methodology, Statistical analysis; Mario P Colombo, Funding acquisition, Writing – original draft; Sabina Sangaletti, Conceptualization, Data curation, Funding acquisition, Investigation, Supervision, Writing – original draft, Writing – review and editing

### Author ORCIDs

Claudio Tripodo http://orcid.org/0000-0002-0821-6231

Sabina Sangaletti http://orcid.org/0000-0001-7047-287X

### Ethics

All experiments involving animals described in this study were approved by the Ministry of Health (authorization numbers 443/2016-PR and 693/2018-PR).

### Decision letter and Author response

Decision letter https://doi.org/10.7554/eLife.69257.sa1
Author response https://doi.org/10.7554/eLife.69257.sa2

## Additional files

### Supplementary files

- Transparent reporting form
- Supplementary file 1. Antibodies for flow cytometry.

### Data availability

All data generated or analyzed during this study are included in the manuscript.

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
