## [Editor Report]

These findings are timely and novel. NPM1-mutated acute myeloid leukemia (AML) is a frequent AML subtype for which new therapeutic approaches are needed. The data presented support the feasibility and the anti-leukemic efficacy of a dendritic cell (DC) vaccine armed with neutrophil extracellular traps (NETs) derived from NPM1-mutated myeloid cells. The new methods presented have important implications and will be of interest to a broad audience from immunology, inflammation and cancer fields.

---

## [Decision Letter]

**Decision letter after peer review:**

Thank you for submitting your article "Neutrophil extracellular traps arm DC vaccination against NPM-mutant myeloproliferation" for consideration by *eLife*. Your article has been reviewed by 2 peer reviewers, and the evaluation has been overseen by a Reviewing Editor and Mone Zaidi as the Senior Editor. The reviewers have opted to remain anonymous.

Essential revisions:

1) This work is novel and may be of interest for hematologists and immunologists. However, some of the claims, particularly those referred to the anti-leukemic activity are not fully supported by the data. Please address the referees comments in the re-submission.

2) Many important points were raised, and the evaluation of the manuscript was meticulous and insightful. Inclusion of a comprehensive response to the comments provided will serve the important purpose of improving the publication and raising the impact of the study, which certainly has merit.

*Reviewer #1 (Recommendations for the authors):*

Please, provide a more detailed description of the experiments in the methods section, particularly for the vaccination protocol.

Some references are reported in brackets while others as superscript. Please revise.

Scale bars should be added to all micrographs.

Figure 1: what is the loading efficiency of NPMc NET and WT NET DCs? May loading efficiency influence their immunogenicity? If yes, please provide loading efficiency data for both DCs.

Figure 1D: please specify in the methods how BM smears have been performed

Figure 1E: CTR is duplicated from a paper published by the same group (Tripodo et al. Cancer Res 2017, Figure 3C, bottom left panel). I recognize this is just a control, but duplication is unacceptable.

Figure 1F-I: the figure legend states "n=5/group", however some groups have less than five replicates and panel H does not show individual replicates. Also, please uniform the style of the histograms all the figures.

Figure 2C: it seems that top and middle panels may be duplicated. If not duplicated it is unclear to this reviewer the meaning of showing these micrographs next to each other. Also, it is hard to read what is written at the top left corner of middle panels. Please clarify.

Figure 3: figure legend should be revised. As it is now, there is no caption for panel D and E.

Figure 3B: labels for x and y axes should be clearer. Please, remove the gating labels from the y axes, as it may be misleading. Also, it would be important to see that WT 45.1 engrafted hematopoietic subpopulations do not show significant abnormalities.

Figure 3C: lower left panel (at least) needs more replicates. There is a wide difference among replicates in WT NET, questioning the actual biological difference between WT NET and NPMc NET. How do the authors explain such marked variability (~0 vs ~40%)? The same holds true for the lower right panel (~50 vs ~100%).

Figure 4: to really capture the actual antileukemic potential of the NET-DC vaccine, experiments with AML cells engrafted in the bone marrow are required.

Figure 4C: this panel is unclear to this reviewer. What does TM mean? What does naïve no TM Vax mean? Please clarify in the figure legend.

Figure 4D: please show the SE or SD of the six replicates. Was the difference between treatment arm significantly different? X axis label is missing.

Figure 4G: can the authors also show OX40 CD8 percent of positive cells together with MFI? This could be added in the supplemental material.

*Reviewer #2 (Recommendations for the authors):*

In this study, the authors achieved their aim to show whether NPMc+ blast-derived NET could be utilized in a DC-based vaccine scheme control myeloproliferation or leukemia outgrowth in mouse models. However, there are some points (below) that need to be addressed for clarity.

Mouse model: Is there a prior characterization of this model? How does the survival of the animal look like in the steady state? Are there any obvious or subtle developmental defects? Is the immune system at the steady state normal? Where does this animal model originate from, Pandolfi group?

Figure 1: NPMc+ NET/DC immunization controls NPMc-driven myeloproliferation

Scale of the histology specimens is missing.

Why is there a difference between the control mouse and the mouse injected with DC loaded with WT NET? Where did the authors obtain the WT NET from? It seems like the DC+WT NET injection, which is basically DCs loaded with MPO and other normal Neutrophil proteins, promotes myeloproliferation, why could this be? Is the control a transgenic mouse too?

Figure 2. Analysis of CD8 T cell frequency and interaction with NPMc+ cells in BM biopsies from control and vaccinated mice.

Any FACS analysis of the BM? What is the %, absolute number and phenotype of CD8^+^ T cells in the BM? Are they more activated, releasing granzyme and maybe IFN-g?

Figure 3: NPMc+ NET/DC immunization selectively impairs NPMc+ mutant hematopoiesis in competitive BMT setting.

This is a very good experiment with extremely promising results, but how do you distinguish NPMc+ 45.1 from the recipient cells that are also 45.1? How can the authors be sure that the BM ablation prior to the BMT is 100% effective?

Also why no CD8 phenotyping was performed by FACS? % and # could be calculated more reliably than histology.

Figure 4: NPMc+ NET vaccination prevents transplantable NPMc+ leukemia cell growth and promotes CD8 lysis.

Route of Leukemia cell injection, i.v.?

Any specific reason why tumor couldn't be transplanted into WT mice that don't have the NPM1 mutation? Have the authors ever tried it? It could give better info about the immune system reaction and vaccine efficacy unless tumor is immediately rejected by WT host. Doing this with WT recipients would also help generalize the effects of NET/DC immunizations for other cancer types. What do authors think about the applicability of this vaccination strategy to solid tumors expressing neoantigens? Is NETosis something that can happen in the any tumor microenvironment?

What is the effect of vaccination on the overall survival of the mice?

CD8 T cell phenotyping FACS plots and gating are required. Why no other CD8 phenotyping is done? Please add if possible.

---

## [Author Response]

Reviewer #1 (Recommendations for the authors):Figure 1: what is the loading efficiency of NPMc NET and WT NET DCs? May loading efficiency influence their immunogenicity? If yes, please provide loading efficiency data for both DCs.

In our original manuscript published in Cancer Research 2017 we reported a superior activation ability of NPMc+ compared to WT NET. Moreover, PMNs isolated from NPMc showed a significantly increased capacity to release NET in response to inflammatory stimuli. In the same paper we first reported that the relocalization of mutant NPMc onto DNA threads license NPM alarmin functions. This explains the superior capacity of mutant NPMc+ NET to load DC, for example with the MPO antigen (Author response image 1). In this figure DC are stained with PKH-26, MPO (in green) is detected through IF analysis.

**Author response image 1. sa2fig1:** 

In the present manuscript we now show that mutant NPMc+NET can load DCs with mutant NPM, which is almost absent in WT NET.

Figure 1D: please specify in the methods how BM smears have been performed

Done. “ BM was flushed with saline from the long bones with a 1 ml-sirynge in a small plate then recovered with the help of forceps and then smeared onto a slide”.

Figure 1E: CTR is duplicated from a paper published by the same group (Tripodo et al. Cancer Res 2017, Figure 3C, bottom left panel). I recognize this is just a control, but duplication is unacceptable.

We amended the Figure 1E accordingly.

Figure 1F-I: the figure legend states "n=5/group", however some groups have less than five replicates and panel H does not show individual replicates. Also, please uniform the style of the histograms all the figures.

All Figures have been revised accordingly. Statistical analyses have been now performed by two additional biostatisticians that have been now listed among the Authors of the manuscript.

Specifically, to overcome the limitation of the number of observations, non-parametric approaches were applied in order to have more robust results.

In addition for some experiments it was possible to increase the number of observations and results obtained from the new analysis adjusted for the different experiments (block variable) have been reported in the manuscript (particularly the new experiments in Figure 4).

We represented individual data as dot plots with the average and appropriate error bars for each group or boxplots. Statistical methods section has been accordingly modified.

We also thank the Reviewer for pointing out the issue regarding the IF panel, which has been replaced with the correct one.

Figure 2C: it seems that top and middle panels may be duplicated. If not duplicated it is unclear to this reviewer the meaning of showing these micrographs next to each other. Also, it is hard to read what is written at the top left corner of middle panels. Please clarify.

The Reviewer probably refer to Figure 2A. The middle and top panel are different as the top panel shows the original figure, whereas the middle and bottom panel show the ouputs of the software segmentation and quantification program.

Figure 3: figure legend should be revised. As it is now, there is no caption for panel D and E.

We revised the figure 3 legend accordingly.

Figure 3B: labels for x and y axes should be clearer. Please, remove the gating labels from the y axes, as it may be misleading. Also, it would be important to see that WT 45.1 engrafted hematopoietic subpopulations do not show significant abnormalities.

We amended Figure 3B, accordingly.

Figure 3C: lower left panel (at least) needs more replicates. There is a wide difference among replicates in WT NET, questioning the actual biological difference between WT NET and NPMc NET. How do the authors explain such marked variability (~0 vs ~40%)? The same holds true for the lower right panel (~50 vs ~100%).

To overcome the limitation of the number of observations, non-parametric approaches were applied in order to have more robust results. Differently from other experiments, such as the ib injection, we are not able to repeat BMT experiment according to the actual law restrictions.

Figure 4: to really capture the actual antileukemic potential of the NET-DC vaccine, experiments with AML cells engrafted in the bone marrow are required.

Done, the new experiment is shown in Figure 4 (panel G-P).

Figure 4C: this panel is unclear to this reviewer. What does TM mean? What does naïve no TM Vax mean? Please clarify in the figure legend.

We apologize for the missing pieces of information. All the abbreviations have been checked and appropriately detailed in the figure legends.

Figure 4D: please show the SE or SD of the six replicates. Was the difference between treatment arm significantly different? X axis label is missing.

All figures now show distribution, mean or median and standard deviation or range.

Figure 4G: can the authors also show OX40 CD8 percent of positive cells together with MFI? This could be added in the supplemental material.

The percentage of OX40+ cells is now included in the new Figure 4I. The figures now include the proper scale bars.

Reviewer #2 (Recommendations for the authors):In this study, the authors achieved their aim to show whether NPMc+ blast-derived NET could be utilized in a DC-based vaccine scheme control myeloproliferation or leukemia outgrowth in mouse models. However, there are some points (below) that need to be addressed for clarity.Mouse model: Is there a prior characterization of this model? How does the survival of the animal look like in the steady state? Are there any obvious or subtle developmental defects? Is the immune system at the steady state normal? Where does this animal model originate from, Pandolfi group?

h-MRP8-NPMc+ transgenic (Tg) mice are phenotypically characterized by the development of an indolent myeloproliferation characterized for the expansion of CD11b+ cells and Gr-1+c-Kit+ myeloblasts without development of overt acute leukemia. They were generated and characterized in the Pandolfi’s lab (doi.org/10.1182/blood-2009-03-208587) and we used this model to show that autoimmunity can worse myeloproliferation in a paper published 2 years ago (Tripodo et al. Cancer Research 2017). Animals are healthy as developing only a mild myeloproliferation. The steady state immune response is normal in these animals.

Figure 1: NPMc+ NET/DC immunization controls NPMc-driven myeloproliferationScale of the histology specimens is missing.Why is there a difference between the control mouse and the mouse injected with DC loaded with WT NET? Where did the authors obtain the WT NET from? It seems like the DC+WT NET injection, which is basically DCs loaded with MPO and other normal Neutrophil proteins, promotes myeloproliferation, why could this be? Is the control a transgenic mouse too?

We apologize with the Reviewer for our lack of clarity. WT NET were generated from WT non-transgenic mice, which do not harbour NPM mutations at the PCR screening. The NET derived from these mice lack NPM cytoplasmatic expression and NPM localization on the DNA threads.

Figure 2. Analysis of CD8 T cell frequency and interaction with NPMc+ cells in BM biopsies from control and vaccinated mice.Any FACS analysis of the BM? What is the %, absolute number and phenotype of CD8^+^ T cells in the BM? Are they more activated, releasing granzyme and maybe IFN-g?

According to Reviewer’ 1 request, we performed additional experiments using the intra-bone injection of leukemic cells, to better mimic the microenvironment of AML. In these experiments we compare the immune microenvironment of mice bearing AML cells either not vaccinated or vaccinated with DC loaded with NPMc+ NET or NPMc-peptides. FASC analysis of flushed BM show that vaccination promotes the activation of CD8 T cells that are proliferating (Ki-67+) and producing TNF and IFNγ. In case of vaccinate mice CD8 T-cells were not exhausted and down-modulating TIM3 and LAG3 (New Figure 4 panel G-L).

The presence of Granzyme B+ cells in contact with NPMc+ cells were detected in situ mice vaccinated with DC loaded with NPMc+ NET, where notably they are in direct contact with NPMc+ cells (Figure 4F).

Figure 3: NPMc+ NET/DC immunization selectively impairs NPMc+ mutant hematopoiesis in competitive BMT setting.This is a very good experiment with extremely promising results, but how do you distinguish NPMc+ 45.1 from the recipient cells that are also 45.1? How can the authors be sure that the BM ablation prior to the BMT is 100% effective?Also why no CD8 phenotyping was performed by FACS? % and # could be calculated more reliably than histology.

The BMT experiment was performed using a 1: CD45.1NPMc+: CD45.2 NPM-wt HSCs. These cells were transplanted into lethally irradiated CD45.1 mice.

The lethal irradiation, as we had the possibility to test and demonstrate along several years of experiments, completely eliminates host myeloid cells (Sangaletti et al. JEM2003, Bassani et al. Front. Immunology 2021). Nonetheless, we agree with the Reviewer about the potential caveat regarding the hypothetical residual host component, which we did not evaluate by FACS. However we reasoned that using CD45.1 mice as recipients could potentially result in an under-estimation of the effect of the NPMc+ vaccine in case of residual host CD45.1. In this experiment we evaluated the local expansion and distribution of CD8^+^ T cells through histopathological analysis. Unfortunately, we could not perform an additional experiment involving lethal irradiation and BMT because of limitations imposed by the animal experimentation project approval. We hope that this will not preclude a positive evaluation of our research efforts.

Figure 4: NPMc+ NET vaccination prevents transplantable NPMc+ leukemia cell growth and promotes CD8 lysis.

In the original version of the study, leukemic cells were subcutaneously injected. We have performed new in vivo experiments using intra-bone injection.

Route of Leukemia cell injection, i.v.?Any specific reason why tumor couldn't be transplanted into WT mice that don't have the NPM1 mutation? Have the authors ever tried it? It could give better info about the immune system reaction and vaccine efficacy unless tumor is immediately rejected by WT host. Doing this with WT recipients would also help generalize the effects of NET/DC immunizations for other cancer types. What do authors think about the applicability of this vaccination strategy to solid tumors expressing neoantigens? Is NETosis something that can happen in the any tumor microenvironment?

We used transgenic mice as we have experience with different transgenic mouse tumor models (NeuT, PyMt) and showed that they are tolerant to tumor-associated antigens, a condition mimicking patients in which cancer arises from pre-malignant conditions. According to the Reviewer’s request, the new in vivo experiments performed using the setting of the intra-bone injection were conducted in non-transgenic C57BL/6 mice as recipients.

Concerning the use of these kind of vaccines in solid tumors, we have preliminary evidence showing the extrusion of traps by tumor cells of epithelial origin (carcinoma cells of both mouse and of human origin. Also in case of tumor traps DNAse treatment, completely destroy the DNA thread

We are using a gp38-antigen model to assess immunogenicity of these tumor traps in mouse models of mammary tumors, also using in vivo DNAse, in a setting in which neutrophils cannot release NET.

What is the effect of vaccination on the overall survival of the mice?

We agree with the Reviewer that an endpoint of the novel immunization strategy that we have proposed and described would be the improvement of survival in leukemic models. In our study we have demonstrated the effects of our experimental immunization on immune activation and control over leukemic progression. Unfortunately, we could not evaluate the survival benefit of our treated and control mouse cohorts as the experimental project didn’t include a formal approval by the Ministry of Health to perform survival studies.

CD8 T cell phenotyping FACS plots and gating are required. Why no other CD8 phenotyping is done? Please add if possible.

Following Reviewer’s suggestion we have better characterized CD8 activation status and reported the results of this characterization in the New Figure 4.